# Successful Management of Recurrent Pyothorax in a Cat: Clinical Findings with Medical and Surgical Approaches

**DOI:** 10.3390/ani15091253

**Published:** 2025-04-29

**Authors:** Hyomi Jang, Seoyeon Kim, Yebeen Lee, Jongwon Park, Hyojun Kwon, Sunyoung Kim, Jiheui Sohn, Jong-in Kim, Dong-In Jung

**Affiliations:** 1VIP Animal Medical Center (Cheongdam), Seoul 06068, Republic of Korea; hm100su@gmail.com (H.J.); syeons@snu.ac.kr (S.K.); dalgom339@naver.com (Y.L.); vetman950314@gmail.com (J.P.); hyojun1220@naver.com (H.K.); duddl430@naver.com (S.K.); wlgml924@naver.com (J.S.); jjongin88@vipah.co.kr (J.-i.K.); 2College of Veterinary Medicine, Gyeongsang National University, Jinju 52828, Republic of Korea

**Keywords:** cat, fibrinolytic agents, pylothorax, thoracotomy, transient myocardial thickening

## Abstract

Pyothorax is a serious condition in cats that involves accumulation of pus in the chest cavity, often requiring urgent treatment. Although most patients respond well to antibiotics and drainage, some can experience recurrence and become more complex. Here, we present a rare case of a cat that initially developed pyothorax and later experienced a recurrence in the form of a paraesophageal abscess. The condition did not respond to medical management alone and ultimately required surgical intervention, including thoracotomy and prolonged pleural lavage, to achieve full recovery. Interestingly, the cat showed a temporary thickening of the heart muscle, which resolved after the infection was controlled. This case highlights the importance of combining medical and surgical treatments in complicated cases of feline pyothorax and suggests that intrathoracic lavage with fibrinolytics may be safe and beneficial.

## 1. Introduction

Feline pyothorax or thoracic empyema is a condition characterized by the accumulation of purulent exudate within the thoracic cavity due to infection [1,2,3]. Typically, cats diagnosed with pyothorax are aged 4–6 years, indicating a relatively younger affected population; however, this condition can manifest in cats of all ages. No breed or sex predispositions have been reported [1,2].

In cats, pyothorax is bilateral in approximately 70–90% of cases and can present as either an acute or chronic disease. Common clinical signs include acute or chronic dyspnea, tachypnea, depression, lethargy, pallor, and anorexia. Physical examination revealed decreased lung sounds, increased abdominal respiratory effort, and paradoxical breathing patterns. Pyrexia is commonly noted upon initial presentation; however, severe sepsis may present with hypothermia and bradycardia [1,4,5,6].

The primary cause of feline pyothorax is bacterial infection, with >80% of the cases involving mixed anaerobic bacterial infections. Known sources of thoracic infection include puncture wounds to the chest wall (such as bite wounds), foreign bodies, esophageal lacerations, and parapneumonic spread [2,3,4,5]. Nonetheless, identifying the definitive etiology of infection remains challenging in most cases (35–67%) [2,7,8].

Currently, there is no standardized therapeutic protocol for the management of pyothorax in veterinary medicine. Generally, treatment involves thoracic drainage and addressing the underlying infectious cause, with surgical intervention being considered if indicated. However, veterinary and human medicine lack substantial data supporting the effectiveness of lavage, tube flushing, suction, and intrapleural fibrinolytic therapy [3,6].

This case report describes a cat initially treated successfully for pyothorax but subsequently found to have developed recurrent caudal mediastinal paraesophageal empyema. Long-term management includes thoracotomy, pleural excision, prolonged chest tube maintenance, and pleural lavage with fibrinolytics. This report discusses these treatment modalities and their associated prognostic considerations.

## 2. Case Description

A 10-year-old male American Curl cat was presented with dyspnea, anorexia, lethargy, and weight loss. Previously, the cat had been diagnosed with chronic kidney disease (IRIS stage 2) and received subcutaneous fluid therapy at home every three days. The cat primarily lived indoors, but occasionally went outside for short supervised walks. The cat was up to date on core vaccinations and had been regularly treated with parasite preventives.

Upon initial examination, severe labored breathing was noted, rectal temperature was 37.4 °C, and systemic systolic blood pressure measured 74 mmHg. Auscultation revealed a grade 4/6 cardiac murmur. Blood tests indicated mild anemia, severe leukocytosis with lymphocytosis, hyperglobulinemia, and a decreased albumin/globulin ratio, with significantly elevated serum concentrations of N-terminal pro-B-type natriuretic peptide (NT-proBNP) and feline serum amyloid A (fSAA) (Table 1).

Thoracic radiography demonstrated a bilateral increase in soft tissue opacity within both sides of the thoracic cavity, obscuring the cardiac silhouette and diaphragmatic outlines. These findings were consistent with bilateral pleural effusion and associated pulmonary atelectasis (Figure 1). Thoracocentesis yielded 100 mL of cloudy, yellowish, sticky fluid (Figure 2A), with a protein concentration of 4.6 mg/dL and a total nucleated cell count of 239.93 K/µL (Table 2). Cytological examination revealed predominantly degenerative neutrophils, macrophages, and lymphocytes along with suspected rod-shaped bacteria (Figure 2B). A septic exudate was suspected based on the cytological findings, and pleural fluid was submitted for bacterial culture (Neodin BioVet Laboratories, Seoul, Republic of Korea) and PCR testing (Idexx Laboratories, Wesbrook, ME, USA).

Because of the heart murmur and elevated NT-proBNP, echocardiography was performed, revealing left ventricular wall thickening (max 7.7 mm), papillary muscle hypertrophy, systolic anterior motion of the mitral valve, mild dynamic left ventricular outflow tract obstruction, mitral and tricuspid valve regurgitation, severe left atrial enlargement (LA:Ao ratio = 2.15), and mild pericardial effusion, consistent with hypertrophic cardiomyopathy phenotype (Figure 3).

Due to the severity of the patient’s condition at presentation and the expected delay in receiving culture results, empirical broad-spectrum antibiotic therapy was initiated. A combination of amoxicillin–clavulanate, metronidazole, and marbofloxacin was selected to provide effective coverage against both aerobic and anaerobic pathogens. Treatment included intravenous administration of amoxicillin–clavulanate (12.5 mg/kg q12h), metronidazole (10 mg/kg q12h), marbofloxacin (2.75 mg/kg q24h), and omeprazole (1 mg/kg q12h). In addition, oral medications consisted of pimobendan (0.25 mg/kg q12h), furosemide (1 mg/kg q12h), and clopidogrel (18.75 mg/cat q24h). Thoracostomy tubes were placed but removed after two days due to pneumothorax.

At this stage, the bacterial culture and PCR results from the pleural fluid collected during the initial thoracocentesis became available. *Fusobacterium russii*, a gram-negative anaerobic bacillus, was isolated and found to be susceptible to all three empirically selected antibiotics. Based on these findings, the initial antimicrobial regimen was maintained without modification. Additionally, the PCR test for feline infectious peritonitis (FIP) was negative.

After eight days of treatment, the cat showed clinical improvement, a reduction in pleural effusion, and improved hematologic parameters. Oral antibiotics were prescribed at discharge (Figure 4).

Two weeks after discharge, antibiotics were discontinued because of severe diarrhea; however, cardiac medications were continued. Regular monitoring was performed, inc luding weekly blood analyses (CBC, biochemistry, blood gas analysis, fSAA, and SDMA), and thoracic and abdominal radiography, as well as abdominal ultrasonography every 1–2 weeks. Two months later, the cat was referred back to our hospital with worsening clinical signs, including cough, dyspnea, and lethargy.

Upon presentation, the body temperature was 41.1 °C with leukocytosis, elevated fSAA, and mild anemia (Table 1).

Radiography revealed increased opacity in the right caudal lung lobe and caudal mediastinum, prompting a CT evaluation, which identified an amorphous lesion with fluid attenuation, gas shadows, and rim enhancement around the caudal vena cava and esophagus, consistent with caudal mediastinal paraesophageal empyema (CMPE) (Figure 5 and Figure 6). Thoracic fluid cytology revealed neutrophils and macrophages without infectious organisms; fluid protein was 4.3 mg/dL, and total nucleated cell count was 21.25 K/µL (Table 2).

As the initial antibiotic therapy—amoxicillin–clavulanate, marbofloxacin, and metronidazole—proved ineffective during the recurrent episode, metronidazole was replaced with meropenem (10 mg/kg IV q8h), based on the previous culture and susceptibility results, to enhance anaerobic coverage and address potential resistance. In addition, meloxicam (0.2 mg/kg SC q24h) and mirtazapine (1.88 mg/cat PO q48h) were initially administered to reduce fever and stimulate appetite, respectively, but both were discontinued as clinical signs improved. Echocardiography indicated the resolution of cardiac abnormalities, allowing the discontinuation of cardiac medications. Clinical deterioration prompted a thoracotomy 12 days after recurrence.

A right intercostal thoracotomy was performed via an approach through the 7th to 8th intercostal space. Following skin and subcutaneous tissue incision, the latissimus dorsi muscle was preserved through dorsal retraction without transection. The scalenus muscle was detached from its attachment at the first rib and subsequently reapproximated at the end of the procedure. The serratus ventralis and intercostal muscles were separated to allow thoracic entry, and a Finochietto retractor was used to maintain surgical exposure.

Thoracotomy revealed severe fibrinous pleuritis, diaphragmatic thickening, and significant inflammatory changes (Figure 7A). The right middle and caudal lung lobes, which appeared darkly discolored, were adhered to each other, and additional adhesions between the diaphragm and lung lobes were identified. These adhesions were surgically dissected and debrided (Figure 7B). The pleural fluid obtained was mildly turbid, yellow, and had low viscosity (Figure 7C). A sample of the fluid was subjected to bacterial culture and antibiotic susceptibility (Neodin BioVet Laboratory, Seoul, Republic of Korea). After evacuation of pleural air, bilateral 12 Fr thoracostomy tubes were placed for long-term management (Figure 7D). Closure of the thoracic wall was initiated by approximating the intercostal muscles using 2-0 polydioxanone (PDS) in a simple interrupted pattern. The thoracic musculature was restored as close to its original position as possible using 4-0 PDS, followed by routine subcutaneous and skin closure. Histopathological examination revealed marked inflammatory infiltrates with neutrophils, macrophages, lymphocytes, plasma cells, fibrin, and cellular debris without neoplasia or active infection (IDEXX Laboratories, Westbrook, ME, USA).

Postoperatively, pleural drainage and lavage were performed for two weeks. After removing as much pleural effusion as possible using bilateral thoracostomy tubes, 20 mL of heparinized Hartmann’s solution (diluted to 10 IU/mL), warmed to body temperature, was instilled and aspirated. This was followed by instillation and aspiration of an additional 20 mL of warm Hartmann’s solution. The study protocol was performed twice daily. Throughout this process, the color and volume of the pleural fluid and degree of fibrin formation were monitored. Following the surgical intervention, the patient’s clinical symptoms rapidly improved. The bacterial culture result from the sample collected intraoperatively during thoracotomy was reported at this time. Based on the results of bacterial culture (*Burkholderia cenocepacia*) and susceptibility testing, antibiotic therapy was unchanged. As no infectious organisms were identified by repeated pleural fluid cytology, all antibiotics, excluding meropenem (10 mg/kg SC q12h), were discontinued. However, owing to the persistent presence of numerous neutrophils in the pleural fluid, prednisolone (0.5 mg/kg PO q12h) and pentoxifylline (10 mg/kg PO q12h) were additionally prescribed, and daily thoracic lavage was continued. By day 29 post-discharge, significant improvement led to the removal of the thoracostomy tubes and cessation of lavage (Table 1). All medications were tapered over the following month.

At the six-month follow-up, the cat showed no recurrence of pyothorax or cardiac abnormalities, maintained weight gain, and remained clinically healthy. However, increased opacity persisted in the right caudal lung lobe (Figure 8).

## 3. Discussion

Numerous case reports on feline pyothorax have indicated a relatively favorable prognosis, with short-term survival rates (hospital discharge) of 72–95% following appropriate treatment and long-term survival (>1 year) of approximately 68%. Recurrence rates are reported to be relatively low, at approximately 6–14% [1,2,3,9].

Presently, the initial presentation included septic exudate due to bacterial infection with *Fusobacterium russii* and impaired cardiac function attributed to the hypertrophic cardiomyopathy (HCM) phenotype, characterized by hypothermia, hypotension, and lethargy. Empirical antibiotic therapy with amoxicillin–clavulanate, marbofloxacin, and metronidazole was promptly initiated along with fluid therapy before culture results, effectively reducing pleural fluid accumulation. Subsequently, antibiotic sensitivity testing confirmed susceptibility, and the initial antibiotics were maintained accordingly. Previous studies have reported successful outcomes with these antibiotics, especially if combined to cover aerobic and anaerobic bacteria, significantly improving survival rates [10].

The exact infection route for *Fusobacterium russii* in this patient remains unclear. The cat had no previous respiratory symptoms, visible trauma, exposure to other animals, or (known bites. *Fusobacterium russii*, a gram-negative anaerobic bacterium, is commonly found in the oral cavity of cats and dogs, often associated with bite injuries [11,12,13,14,15]. Considering that the patient’s chronic kidney disease management involved repeated subcutaneous fluid administration and cat grooming behavior, skin penetration by needles and subsequent grooming may have facilitated the infection. Although subcutaneous fluid therapy is widely used in the management of feline chronic kidney disease with mild complications such as fluid overload and localized injection site reactions, such as infection, edema, or inflammation, there have been no previous reports directly linking this treatment to the development of pyothorax. Thus, this case highlights the need for awareness of potentially severe complications associated with subcutaneous fluid therapy, such as pyothorax.

At the time of recurrence, the clinical presentation had progressed to caudal mediastinal paraesophageal empyema (CMPE), which differed from the initial acute and severe septic pyothorax that had responded rapidly to antibiotic therapy. Despite the isolation of *Burkholderia cenocepacia* on culture and its demonstrated susceptibility to the administered antibiotics, the patient’s clinical condition continued to deteriorate. In contrast, previous studies have reported successful management of recurrent feline pyothorax using the same antibiotics employed in the initial treatment, highlighting a discrepancy with the present case [3]. In contrast, studies in dogs have reported poor prognosis and high mortality rates associated with recurrence [16].

Given that no other infectious organisms were identified through cytological or histopathological examination, and that even the cultured bacteria showed in vitro susceptibility to the treatment regimen with minimal clinical improvement, it was considered unlikely that the recurrence was due to insufficient antibiotic coverage during the initial episode.

There are several possible explanations for this. *B. cenocepacia*, although identified during the recurrent episode, was not likely to be the primary cause of the pyothorax. It is typically considered an environmental pathogen and has recently been implicated in nosocomial infections in cats [17]. Provided these characteristics, *B. cenocepacia* was more likely to be a contaminant introduced during sample collection.

If the second episode resulted from treatment failure of the initial infection, this possibility appears low. There are no previous reports of CMPE in cats caused by *Fusobacterium russii*; however, infections involving *Fusobacterium nucleatum* in other feline cases responded well to thoracostomy and antibiotic therapy, without requiring surgical intervention [18,19].

Moreover, the presence of a difficult-to-culture bacterial infection or previous antibiotic treatment at another clinic may have partially sterilized the infection, resulting in negative culture results. Another report involving Nocardia spp. infection in dogs described the involvement of the diaphragm and caudal lung lobe, similar to our case [20]. However, this was considered less likely, given the consistent absence of organisms on repeated cytological and histopathological examinations in this patient.

A previous study on CMPE in five dogs suggested foreign body-induced mesothelial irritation or bronchopneumonia as triggers [21]. Here, no infectious organisms or evidence of esophageal perforation were identified.

In the treatment of CMPE, systemic antibiotic administration in combination with surgical exploration or drainage is generally recommended [19,21,22]. Similarly, the cat’s condition rapidly improved after thoracotomy, pleural debridement, and thoracostomy tube placement. As reported in previous studies, thoracotomy combined with extended thoracic lavage with fibrinolytics are effective in cases where simple thoracocentesis fails to sufficiently evacuate the exudate and inflammatory products, thereby justifying the use of surgical intervention, thoracostomy tube placement, and repeated lavage [4,6,9,23]. The efficacy of fibrinolytic agents in feline thoracic lavage remains unclear, although their use has been reported in human medicine [24,25,26,27,28]. A previous study on dogs with pyothorax reported improved short-term survival following the addition of heparin (10 U/mL) to warm crystalloid lavage solutions [16]. Another study proposed the clinical potential of heparinized lavage fluid (1500 U/100 mL) [23]. However, there is currently no evidence supporting the use of such protocols in feline patients [6]. In the present case, 42 days of drainage and lavage via a thoracostomy tube using a heparinized solution resulted in no severe adverse effects other than mild pneumothorax, subcutaneous emphysema, and tube dislocation, suggesting potential benefits for the long-term management of feline pyothorax.

Although an esophagoscopy was considered to rule out esophageal perforation or foreign body migration, it was not performed in this case. The patient exhibited no upper gastrointestinal signs such as vomiting, and computed tomography as well as intraoperative findings did not reveal any foreign material. Additionally, histopathological examination showed only inflammatory cell infiltration without evidence of foreign bodies or microorganisms, suggesting a low likelihood of esophageal involvement.

The reversible thickening of the left ventricular wall in this cat was diagnosed as transient myocardial thickening (TMT), which resolved completely within two months of treatment. TMT, distinct from irreversible hypertrophic cardiomyopathy, is characterized by reversible myocardial hypertrophy secondary to myocardial edema or inflammation [29,30,31,32]. TMT in cats has been linked to infections (*Bartonella henselae*, *Streptococcus canis*, *Toxoplasma gondii*, feline immunodeficiency virus, feline coronavirus, and others), and stress-related events, such as anesthetic procedures and surgery [33,34,35,36,37,38,39,40,41]. *Fusobacterium* spp. have previously been associated with pericarditis and effusion in cats [42]. To our knowledge, this is the first reported case describing concurrent pyothorax and TMT. One limitation of this case is that serum cardiac troponin I was not measured during the initial cardiac evaluation. Although NT-proBNP levels, thoracic radiography, and echocardiography were utilized to assess cardiac function, the absence of troponin data may limit the comprehensive evaluation of myocardial injury. However, close follow-up with serial imaging over six months confirmed complete resolution of cardiac abnormalities.

## 4. Conclusions

This case report documents a cat initially treated successfully for bacterial pyothorax and TMT, which subsequently recurred as a pyogranulomatous empyema that was unresponsive to antibiotics alone. Surgical interventions, including thoracotomy and long-term thoracostomy tube drainage with fibrinolytic lavage, resulted in complete resolution.

## Figures and Tables

**Figure 1 animals-15-01253-f001:**
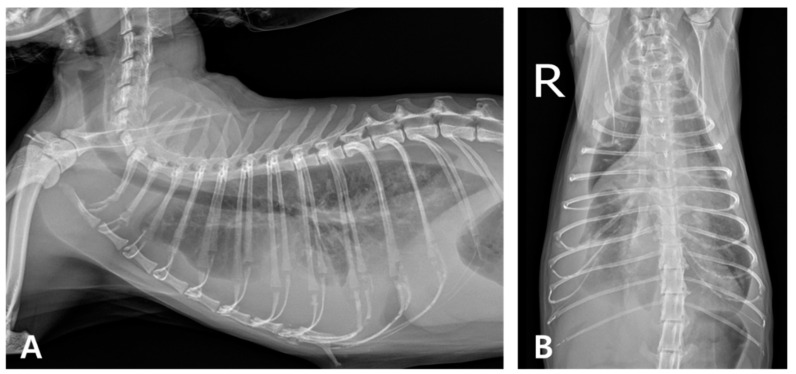
Thoracic radiography at the initial onset. (**A**) Right lateral and (**B**) ventrodorsal views (R: right). Marked bilateral soft tissue opacity is observed within the thoracic cavities, obscuring both the cardiac silhouette and diaphragmatic margins. The lung lobes are retracted bilaterally from the thoracic walls, consistent with bilateral pleural effusion and associated atelectasis.

**Figure 2 animals-15-01253-f002:**
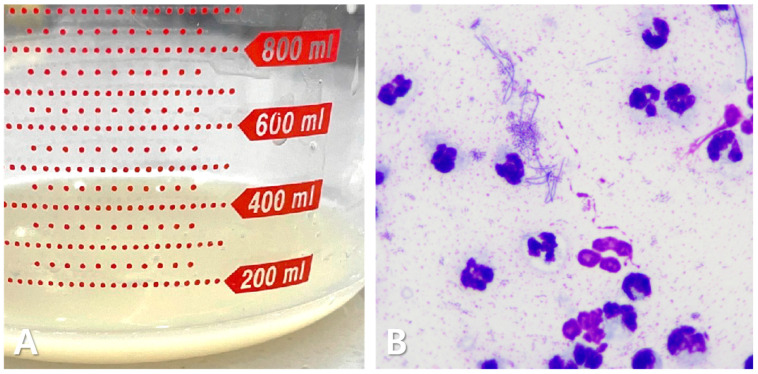
Pleural effusion in the cat. Cloudy, yellowish, and sticky fluid (**A**). Cytological examination shows a predominance of degenerative neutrophils, with some macrophages and lymphocytes. Numerous rod-shaped bacterial structures are present (**B**).

**Figure 3 animals-15-01253-f003:**
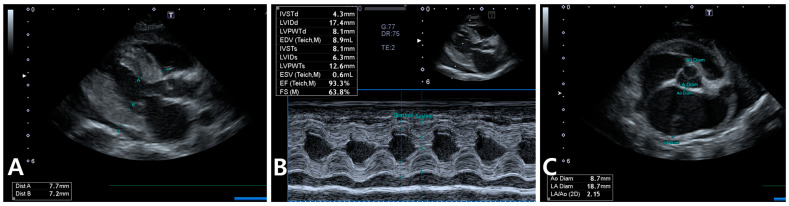
Echocardiography in the cat shows a maximum left ventricular wall thickness of approximately 7.7 mm during diastole, along with papillary muscle hypertrophy, systolic anterior motion of the mitral valve, and mild pericardial effusion (**A**). Similar findings are observed in M-mode imaging (**B**). Additionally, severe left atrial enlargement is evident (**C**).

**Figure 4 animals-15-01253-f004:**
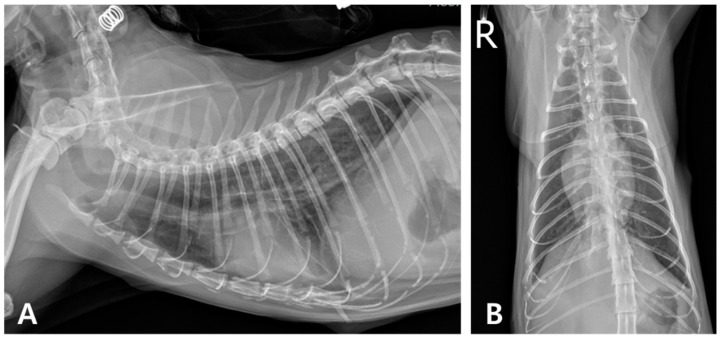
Thoracic radiography on day 8. (**A**) Right lateral radiographic and (**B**) ventrodorsal views (R: right) show a significant reduction in pleural fluid within the thoracic cavity.

**Figure 5 animals-15-01253-f005:**
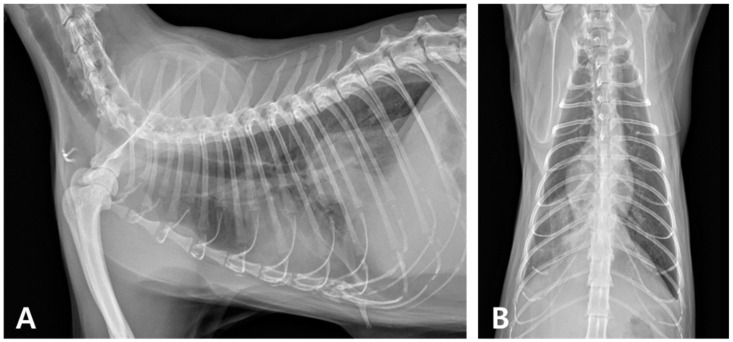
Thoracic radiography during recurrence. (**A**) Right lateral radiographic and (**B**) ventrodorsal views reveal increased opacity in the right caudal lung lobe, along with a soft tissue opaque structure in the caudal mediastinum.

**Figure 6 animals-15-01253-f006:**
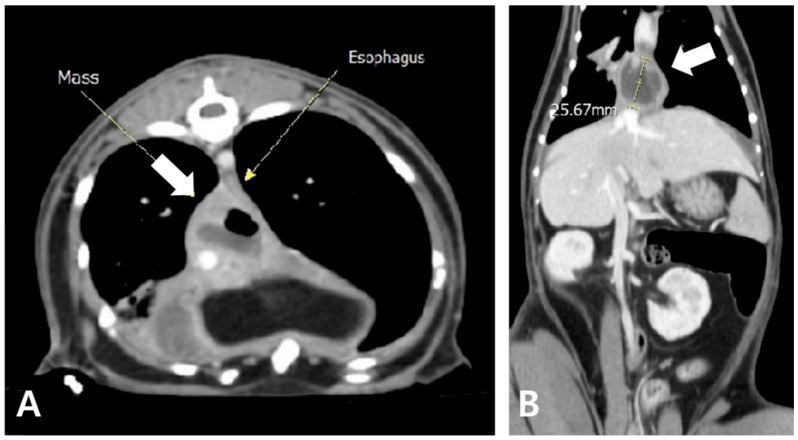
Axial (**A**) and coronal view (**B**) of post-contrast CT findings. An amorphous mass-like structure (white arrow) is observed near the caudal mediastinum, located to the right and dorsal to the esophagus. The lesion exhibits fluid attenuation with areas of gas opacity and demonstrates rim enhancement following contrast administration.

**Figure 7 animals-15-01253-f007:**
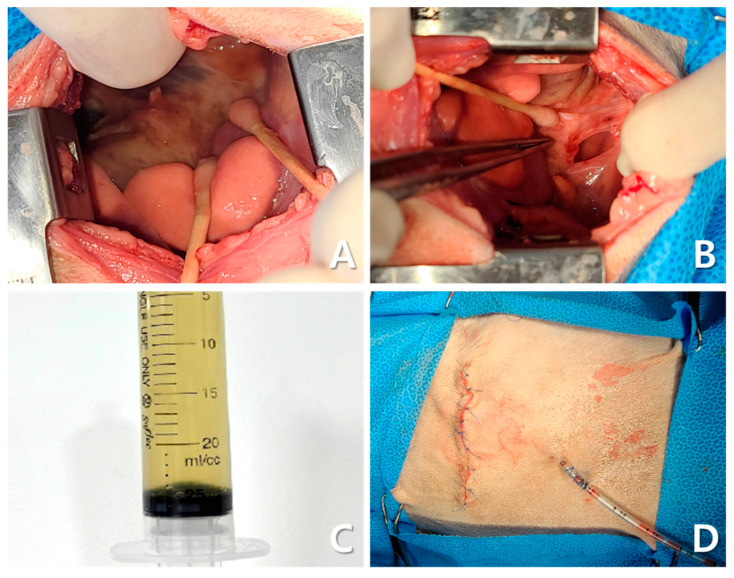
Intraoperative findings during thoracotomy. Diffuse fibrinous pleuritis, diaphragmatic thickening, and inflammatory changes are observed in the right thoracic cavity (**A**). Owing to adhesion between the right middle and caudal lung lobes and the diaphragm, debridement and separation were performed (**B**). Yellow-tinged pleural fluid with low viscosity and slightly increased turbidity is observed (**C**). A bilateral thoracostomy tube was placed for long-term drainage (**D**).

**Figure 8 animals-15-01253-f008:**
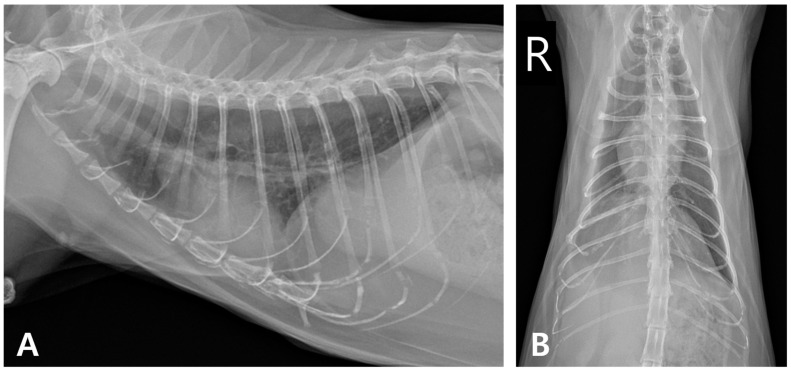
Thoracic radiographic monitoring after discontinuation of medication. (**A**) Right lateral radiographic and (**B**) ventrodorsal views (R: right).

**Table 1 animals-15-01253-t001:** Hematological parameters of the cat at initial onset, recurrence, and thoracostomy tube removal day.

	Initial Onset	Recurrent Onset	Tube Removal	2 Weeks After Cessation of Lavage	2 Months Post-Medication Withdrawal	1 Year Post-Treatment	Reference Range
Hematocrit (%)	22	26.8	32.7	33.4	36.7	48.4	30.3–52.3
White blood cells (k/μL)	51.98	21.92	6.98	6.91	6.44	6.34	2.87–17.02
Neutrophils (k/μL)	0.45	12.3	2.82	4.41	2.38	2.43	2.30–10.29
Lymphocytes (k/μL)	37.43	8.98	2.67	1.71	1.56	2.01	0.92–6.88
Monocytes (k/μL)	13.84	0.44	0.58	0.43	0.18	0.16	0.05–0.67
Eosinophils (k/μL)	0.07	0.06	0.87	0.29	1.97	1.48	0.17–1.57
Platelet (k/μL)	386	239	456	371	270	282	151–600
Total protein (g/dL)	7.7	7.0	-	-	7.1	7.0	5.7–8.9
Albumin (g/dL)	2.4	2.6	-	-	3.1	2.8	2.2–4.0
Globulin (g/dL)	5.3	4.4	-	-	4.0	4.2	2.8–5.1
A:G ratio	0.4	0.6	-	-	0.8	0.6	
BUN (mg/dL)	43	26	-	49	37	36	9–29
Creatinine (mg/dL)	0.8	1.2	-	1.0	1.9	1.9	0.8–2.4
SDMA (μg/dL)	19	20	4	10	-	16	0–14
fSAA (mg/mL)	24.9	17.3	<5	<5	<5	<5	<5
NT-proBNP (μg/dL)	1014.8	122	-	-	<50	53.7	<100

A:G, Albumin-to-globulin; BUN = Blood urea nitrogen, SDMA = Symmetric dimethylarginine; fSAA, Feline serum amyloid A, NT-proBNP = N-terminal pro-B-type natriuretic peptide.

**Table 2 animals-15-01253-t002:** Pleural effusion analysis in the cat.

	Initial Onset	Recurrent Onset
Total nucleated cell count (K/μL)	239.93	21.25
Total protein (g/dL)	4.6	4.3
Triglyceride (mg/dL)	<10	<10
Total cholesterol (mg/dL)	74	106

## Data Availability

The original contributions of this study are presented. Further inquiries can be directed to the corresponding author.

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
