# Peer review of "Successful Management of Recurrent Pyothorax in a Cat: Clinical Findings with Medical and Surgical Approaches"

_animals, 2025, doi:10.3390/ani15091253_

Round 1
Reviewer 1 Report
Comments and Suggestions for Authors
Manuscript with ID: animals- 3585576 and Title: Successful Management of Recurrent Pyothorax in a Cat: Clinical Findings with Medical and Surgical Approaches
This report presents a case of medical and surgical management of recurrent pyothorax in a 10-year-old cat, highlighting the clinical findings, diagnostic assessments, and treatment strategies.
This article deals with a very interesting topic for clinical practice and I appreciate the work done by the authors. However, I do feel that discussion and conclusions have some issues that have to be addressed.
Comments:
Was the pyothorax bilateral or unilateral?
Page 2, line 57: please delete the word ‘’simple’’
Page 2, line 63: was presented
Page 5, line 120-122: provide a more thorough description of the follow up of the patient regarding the pleural effusion (were any diagnostic imaging procedures performed?) and the physical and laboratory examinations. Two weeks is a short period for antibiotic discontinuation in cases of pyothorax.
Page 6, line 146: please provide information regarding the thoracotomy (approach).
Page 6, line 148-150: so surgery was done for diagnosis and thoracostomy tube placement? Please provide a more detailed description.
Page 9, line 238: why wasn’t an esophagoscopy performed?
Page 9, line 240: please give more information regarding the indications for surgical intervention in cats with pyothorax
Page 9, line 244: ‘’The efficacy of fibrinolytic agents in feline thoracic lavage remains’’. Please give more information regarding the use of fibrinolytic agents in feline pyothorax.
Page 9, line 245: ‘’although they are commonly used in human medicine’’ there are papers stating the contrary. Please rephrase.
Page 9, line 246: ‘’42-day thoracostomy’’ please comment on this prolonged period of thoracostomy tubes.
Page 10-12, references: please correct as they are double-numbered
Author Response
Comments to the Author
This report presents a case of medical and surgical management of recurrent pyothorax in a 10-year-old cat, highlighting the clinical findings, diagnostic assessments, and treatment strategies.
This article deals with a very interesting topic for clinical practice and I appreciate the work done by the authors. However, I do feel that discussion and conclusions have some issues that have to be addressed.
1) Was the pyothorax bilateral or unilateral?
Response:
Thank you for this important question. In this case, the cat was diagnosed with bilateral pyothorax at initial presentation, which is consistent with previously reported findings in feline patients. To improve clarity, we have revised the description of the thoracic radiographs in lines 78–81 and elaborated on the bilateral nature of the pleural effusion in the legend of Figure 1 in the revised manuscript.
Line 79-82
Thoracic radiography demonstrated a bilateral increase in soft tissue opacity within both sides of the thoracic cavity, obscuring the cardiac silhouette and diaphragmatic outlines. These findings were consistent with bilateral pleural effusion and associated pulmonary atelectasis (Figure 1).
Line 92-95
Figure 1. Thoracic radiography at the initial onset. (A) Right lateral and (B) ventrodorsal views. Marked bilateral soft tissue opacity is observed within the thoracic cavities, obscuring both the cardiac silhouette and diaphragmatic margins. The lung lobes are retracted bilaterally from the thoracic walls, consistent with bilateral pleural effusion and associated atelectasis.
2) Page 2, line 57: please delete the word ‘’simple’’
Response:
Thank you for pointing this out. We have removed the word “simple” from the sentence as recommended to avoid any misinterpretation.
3) Page 2, line 63: was presented
Response:
Thank you for your suggestion. We have revised the sentence to use “was presented” as recommended.
4) Page 5, line 120-122: provide a more thorough description of the follow up of the patient regarding the pleural effusion (were any diagnostic imaging procedures performed?) and the physical and laboratory examinations. Two weeks is a short period for antibiotic discontinuation in cases of pyothorax.
Response:
Thank you for this insightful comment. We agree that additional detail on the follow-up period is important. In total, the patient received antibiotic therapy for approximately 22 days, including 8 days of inpatient intravenous antibiotics and 14 days of oral antibiotics at home. However, oral antibiotics were discontinued earlier than planned due to the development of severe diarrhea, at the owner’s request. In the revised manuscript, we have clarified that the term “gastrointestinal side effect” has been updated to “severe diarrhea” for accuracy.
Additionally, we have expanded the description of the follow-up process, including regular physical examination, thoracic imaging, and hematologic monitoring to assess for signs of recurrence after antibiotic discontinuation.
Line 133
Two weeks after discharge, antibiotics were discontinued because of severe diarrhea;
5) Page 6, line 146: please provide information regarding the thoracotomy (approach).
Response:
Thank you for your valuable comment. In response, we have expanded the description of the thoracotomy procedure in the revised manuscript. Specifically, we have added details regarding the surgical approach as follows.
Line 169-186
A right intercostal thoracotomy was performed via an approach through the 7th to 8th intercostal space. Following skin and subcutaneous tissue incision, the latissimus dorsi muscle was preserved through dorsal retraction without transection. The scalenus muscle was detached from its attachment at the first rib and subsequently reapproximated at the end of the procedure. The serratus ventralis and intercostal muscles were separated to allow thoracic entry, and a Finochietto retractor was used to maintain surgical exposure.
Thoracotomy revealed severe fibrinous pleuritis, diaphragmatic thickening, and significant inflammatory changes (Figure 7A). The right middle and caudal lung lobes, which appeared darkly discolored, were adhered to each other, and additional adhesions between the diaphragm and lung lobes were identified. These adhesions were surgically dissected and debrided (Figure 7B). The pleural fluid obtained was mildly turbid, yellow, and had low viscosity (Figure 7C). A sample of the fluid was subjected to bacterial culture and antibiotic susceptibility. (Neodin BioVet. Laboratory, Seoul, Korea). After evacuation of pleural air, bilateral 12 Fr thoracostomy tubes were placed for long-term management (Figure 7D). Closure of the thoracic wall was initiated by approximating the intercostal muscles using 2-0 polydioxanone (PDS) in a simple interrupted pattern. The thoracic musculature was restored as close to its original position as possible using 4-0 PDS, followed by routine subcutaneous and skin closure.
6) Page 6, line 148-150: so surgery was done for diagnosis and thoracostomy tube placement? Please provide a more detailed description.
Response:
Thank you for your helpful comment. We have clarified that the thoracotomy was performed not only for diagnostic purposes—such as direct visualization of the lesion and tissue sampling—but also for therapeutic purposes. These included the surgical separation of adhesions between the lung lobes and diaphragm, as well as the placement of bilateral thoracostomy tubes. The revised manuscript now includes a more detailed account of these surgical interventions on lines 169–186.
7) Page 9, line 238: why wasn’t an esophagoscopy performed?
Response:
Thank you for raising this important point. In this case, the patient did not exhibit upper gastrointestinal signs such as vomiting prior to the development of CMPE. Additionally, computed tomography and intraoperative findings revealed no evidence of foreign bodies, and histopathological examination identified only inflammatory cells without any foreign material or microorganisms. Therefore, the likelihood of esophageal perforation due to a foreign body was considered low, and esophagoscopy was not pursued. This rationale has been added to the discussion section (page 9, line 272-278).
Line 297-303
Although an esophagoscopy was considered to rule out esophageal perforation or foreign body migration, it was not performed in this case. The patient exhibited no upper gastrointestinal signs such as vomiting, and computed tomography as well as intraoperative findings did not reveal any foreign material. Additionally, histopathological examination showed only inflammatory cell infiltration without evidence of foreign bodies or microorganisms, suggesting a low likelihood of esophageal involvement.
8) Page 9, line 240: please give more information regarding the indications for surgical intervention in cats with pyothorax
Response:
Thank you for your insightful comment. As suggested, we have added information regarding the indications for surgical intervention in feline pyothorax. Specifically, we stated that surgical exploration should be considered when thoracocentesis alone fails to adequately remove pleural fluid and inflammatory debris, or when repeated lavage is necessary. This addition can be found in the discussion section (page 9, line 255-261).
Line 280-286
In the treatment of CMPE, systemic antibiotic administration in combination with surgical exploration or drainage is generally recommended [12,14,15]. Similarly, the cat’s condition rapidly improved after thoracotomy, pleural debridement, and thoracostomy tube placement. As reported in previous studies, thoracotomy combined with extended thoracic lavage with fibrinolytics are effective in cases where simple thoracocentesis fails to sufficiently evacuate the exudate and inflammatory products, thereby justifying the use of surgical intervention, thoracostomy tube placement, and repeated lavage. [4,6,16,30].
9) Page 9, line 244: ‘’The efficacy of fibrinolytic agents in feline thoracic lavage remains’’. Please give more information regarding the use of fibrinolytic agents in feline pyothorax.
Response:
Thank you for this valuable suggestion. We have expanded the discussion to include available information on the use of fibrinolytic agents in veterinary medicine. While there are currently no published studies specifically evaluating the use of fibrinolytics in cats with pyothorax, limited case reports and experimental studies have described their application in dogs and other veterinary species. These examples, along with their potential benefits and limitations, have been added to the discussion section.
Line 287-296
The efficacy of fibrinolytic agents in feline thoracic lavage remains unclear, although their use has been reported in human medicine [17–21]. A previous study on dogs with pyothorax reported improved short-term survival following the addition of heparin (10 U/mL) to warm crystalloid lavage solutions [9]. Another study proposed the clinical potential of heparinized lavage fluid (1500 U/100 mL) [16]. However, there is currently no evidence supporting the use of such protocols in feline patients [6]. In the present case, 42 days of drainage and lavage via a thoracostomy tube using a heparinized solution resulted in no severe adverse effects other than mild pneumothorax, subcutaneous emphysema, and tube dislocation, suggesting potential benefits for the long-term management of feline pyothorax.
10) Page 9, line 245: ‘’although they are commonly used in human medicine’’ there are papers stating the contrary. Please rephrase.
Response:
Thank you for these helpful suggestions. We revised as reviewer’s comment.
Line 287-288
The efficacy of fibrinolytic agents in feline thoracic lavage remains unclear, although their use has been reported in human medicine [17–21].
11) Page 9, line 246: ‘’42-day thoracostomy’’ please comment on this prolonged period of thoracostomy tubes.
Response:
Thank you for these helpful suggestions. We revised as reviewer’s comment.
Line 292-296
In the present case, 42 days of drainage and lavage via a thoracostomy tube using a heparinized solution resulted in no severe adverse effects other than mild pneumothorax, subcutaneous emphysema, and tube dislocation, suggesting potential benefits for the long-term management of feline pyothorax.
11) Page 10-12, references: please correct as they are double-numbered
Response:
Thank you for these helpful suggestions. We have also carefully reviewed the reference list and corrected the numbering to eliminate duplication and ensure consistency with MDPI formatting.

Reviewer 2 Report
Comments and Suggestions for Authors
Limits of the study:
- not having investigated cardiac troponin
- not having re-evaluated blood tests after treatment (antibiotics, furosemide etc in a CKD cat)
It's not clear why they treat the cat with subcutaneous fluid therapy for CKD stage 2, but this isn't an authors' fault.
I think the work was well done and well argued.
The work is interesting. I have never used Hartamann's solution for lavage, but we usually use sterile saline (NaCl). I would have argued better the choice of using this solution with some bibliography.
Author Response
Reviewer 2
1) Not having investigated cardiac troponin.
Response:
Thank you for pointing out this important limitation. In this case, cardiac assessment was performed using NT-proBNP measurement, thoracic radiography, and echocardiography. Unfortunately, serum cardiac troponin I was not evaluated due to resource constraints at the time of initial presentation. However, as noted in the final paragraph of the case report, the patient was closely monitored for six months following recovery through repeated thoracic radiographs and echocardiography, which confirmed the resolution of cardiac abnormalities. We have acknowledged the absence of troponin I testing as a limitation in the discussion section.
Line 314-319
One limitation of this case is that serum cardiac troponin I was not measured during the initial cardiac evaluation. Although NT-proBNP levels, thoracic radiography, and echocardiography were utilized to assess cardiac function, the absence of troponin data may limit the comprehensive evaluation of myocardial injury. However, close follow-up with serial imaging over six months confirmed complete resolution of cardiac abnormalities.
2) Not having re-evaluated blood tests after treatment (antibiotics, furosemide etc in a CKD cat).
Response:
Thank you for your thoughtful comment. As noted in Table 1 of the original manuscript, bloodwork was performed at the time of thoracostomy tube removal, which occurred more than one month after discontinuation of furosemide and other cardiac medications (see line 149: “allowing the discontinuation of cardiac medications”). Following your advice, we have now supplemented Table 1 with additional hematological data collected two weeks after tube removal and after complete cessation of medications. These additions provide further confirmation of the patient’s stable clinical condition and recovery.
Line 75-78
Table 1. Hematological parameters of the cat at initial onset, recurrence, and thoracostomy tube removal day.
|
|
Initial onset |
Recurrent onset |
Tube removal |
2 weeks after cessation of lavage |
2 months post-medication withdrawal |
1-year post-treatment |
Reference range |
|
Hematocrit (%) |
22 |
26.8 |
32.7 |
33.4 |
36.7 |
48.4 |
30.3–52.3 |
|
White Blood Cells (k/μL) |
51.98 |
21.92 |
6.98 |
6.91 |
6.44 |
6.34 |
2.87–17.02 |
|
Neutrophils (k/μL) |
0.45 |
12.3 |
2.82 |
4.41 |
2.38 |
2.43 |
2.30–10.29 |
|
Lymphocytes (k/μL) |
37.43 |
8.98 |
2.67 |
1.71 |
1.56 |
2.01 |
0.92–6.88 |
|
Monocytes (k/μL) |
13.84 |
0.44 |
0.58 |
0.43 |
0.18 |
0.16 |
0.05–0.67 |
|
Eosinophils (k/μL) |
0.07 |
0.06 |
0.87 |
0.29 |
1.97 |
1.48 |
0.17–1.57 |
|
Platelet (k/μL) |
386 |
239 |
456 |
371 |
270 |
282 |
151–600 |
|
Total protein (g/dL) |
7.7 |
7.0 |
- |
- |
7.1 |
7.0 |
5.7–8.9 |
|
Albumin (g/dL) |
2.4 |
2.6 |
- |
- |
3.1 |
2.8 |
2.2–4.0 |
|
Globulin (g/dL) |
5.3 |
4.4 |
- |
- |
4.0 |
4.2 |
2.8–5.1 |
|
A:G ratio |
0.4 |
0.6 |
- |
- |
0.8 |
0.6 |
|
|
BUN (mg/dL) |
43 |
26 |
- |
49 |
37 |
36 |
9–29 |
|
Creatinine (mg/dL) |
0.8 |
1.2 |
- |
1.0 |
1.9 |
1.9 |
0.8–2.4 |
|
SDMA (μg/dL) |
19 |
20 |
4 |
10 |
- |
16 |
0–14 |
|
fSAA (mg/mL) |
24.9 |
17.3 |
<5 |
<5 |
<5 |
<5 |
<5 |
|
NT-proBNP (μg/dL) |
1014.8 |
122 |
- |
- |
<50 |
53.7 |
<100 |
A:G, albumin-to-globulin; BUN = Blood Urea Nitrogen, SDMA = Symmetric Dimethylarginine; fSAA, Feline Serum Amyloid A, NT-proBNP = N-terminal pro-B-type Natriuretic Peptide
3) It's not clear why they treat the cat with subcutaneous fluid therapy for CKD stage 2, but this isn't an authors' fault.
I think the work was well done and well argued.
The work is interesting. I have never used Hartamann's solution for lavage, but we usually use sterile saline (NaCl). I would have argued better the choice of using this solution with some bibliography.
Response:
Thank you for this valuable comment. We appreciate the opportunity to elaborate on our choice. In our clinical experience, we commonly use balanced crystalloid solutions such as lactated Ringer’s solution (Hartmann’s) for intrathoracic lavage due to their physiological compatibility and buffering properties. Most veterinary literature, including standard references, recommends 0.9% NaCl or balanced solutions like LRS for pleural lavage. Specifically, reference [6] (The Cat, Elsevier Health Sciences, 2024), in FIG. 33.34 (“Use of a thoracostomy tube in a cat”), acknowledges the use of balanced crystalloids in this context.
Additionally, as noted earlier, this cat had been receiving unsupervised subcutaneous fluid therapy with an unconfirmed protocol prior to referral, based on non-veterinary sources. Following this episode, the owner became more cautious and discontinued unapproved fluid administration. We have incorporated these clarifications into the revised discussion section.

Reviewer 3 Report
Comments and Suggestions for Authors
Overall, the document is well-written, but it is necessary to clarify the timeline of events.
- Line 12: The sentence is unclear; please rewrite it.
- Line 67: The sentence is unclear; please rewrite it.
- Lines 83-84: A septic exudate was suspected. I assume the sample was sent to the laboratory for culture. If this was done, it should be indicated here.
- Lines 108-110: How was the decision made to use those three antibiotics? Please specify.
- Line 113: The sentence is generic, and it is unclear when the culture sample was collected.
- Line 115: Which antibiotics were used? All three? Please specify.
- Line 123: Which antibiotics were used, and for how many days?
- Line 143: Which antibiotic therapy are you referring to? It is unclear. Also, on what basis was meropenem chosen?
- Line 150: Was a culture performed on the pleural effusion?
- Line 170: When was this culture performed and on which sample? It is unclear; please rewrite.
- Line 174: So, was corticosteroid added while NSAIDs were still being administered? This is unclear.
I would also like to know the bacterial count of the cultures.
In the discussion, the hypothesis that a 3-week antibiotic treatment might have been insufficient—which could have contributed to the formation of the empyema—was not considered.
• Line 251: It states that ventricular thickening resolved after 2 months, but in line 179, it is mentioned that the follow-up was done after 6 months. Please rewrite to clarify the timeline.
Author Response
Reviewer 3
Overall, the document is well-written, but it is necessary to clarify the timeline of events.
- Line 12: The sentence is unclear; please rewrite it.
Response:
We revised the sentence in line 12 of the Simple Summary to improve clarity. The new version clearly states that the cat did not respond to antibiotics alone and required surgical treatment, which led to full recovery.
Line 11-15
Here, we present a rare case of a cat that initially developed pyothorax and later experienced a recurrence in the form of a paraesophageal abscess. The condition did not respond to medical management alone and ultimately required surgical intervention, including thoracotomy and prolonged pleural lavage, to achieve full recovery.
- Line 67: The sentence is unclear; please rewrite it.
Response:
The sentence was rewritten as follows.
Line 67-68
The cat was up to date on core vaccinations and had been regularly treated with parasite preventives.
- Lines 83-84: A septic exudate was suspected. I assume the sample was sent to the laboratory for culture. If this was done, it should be indicated here.
Response:
Yes, the pleural fluid obtained via thoracocentesis was submitted for aerobic and anaerobic culture. This detail has been added in the revised manuscript.
Line 86-88
A septic exudate was suspected based on cytological findings, and pleural fluid was submitted for bacterial culture (Neodin BioVet Laboratories, Seoul, Republic of Korea) and PCR testing (Idexx Laboratories, Wesbrook, ME, USA).
- Lines 108-110: How was the decision made to use those three antibiotics? Please specify.
- Line 113: The sentence is generic, and it is unclear when the culture sample was collected.
- Line 115: Which antibiotics were used? All three? Please specify.
Response to Questions 4 &5&6:
Thank you for these important questions. The initial antibiotic combination—amoxicillin-clavulanate, metronidazole, and marbofloxacin—was selected empirically due to the severity of the patient's clinical condition. As bacterial culture and susceptibility testing require several days, we chose a broad-spectrum and aggressive approach to provide immediate coverage for both aerobic and anaerobic pathogens. Pleural fluid was collected at the time of initial thoracocentesis and submitted to Laboratories for aerobic and anaerobic culture and PCR. Once the culture results confirmed Fusobacterium russii, and susceptibility testing showed that the selected antibiotics were appropriate, the same regimen was maintained.
Line 112-129
Due to the severity of the patient's condition at presentation and the expected delay in receiving culture results, empirical broad-spectrum antibiotic therapy was initiated. A combination of amoxicillin-clavulanate, metronidazole, and marbofloxacin was selected to provide effective coverage against both aerobic and anaerobic pathogens. Treatment included intravenous administration of amoxicillin-clavulanate (12.5 mg/kg q12h), metronidazole (10 mg/kg q12h), marbofloxacin (2.75 mg/kg q24h), and omeprazole (1 mg/kg q12h). In addition, oral medications consisted of pimobendan (0.25 mg/kg q12h), furosemide (1 mg/kg q12h), and clopidogrel (18.75 mg/cat q24h). Thoracostomy tubes were placed but removed after two days due to pneumothorax.
At this stage, the bacterial culture and PCR results from the pleural fluid collected during the initial thoracocentesis became available. Fusobacterium russii, a gram-negative anaerobic bacillus, was isolated and found to be susceptible to all three empirically selected antibiotics. Based on these findings, the initial antimicrobial regimen was maintained without modification. Additionally, the PCR test for feline infectious peritonitis (FIP) was negative.
After eight days of treatment, the cat showed clinical improvement, a reduction in pleural effusion, and improved hematologic parameters. Oral antibiotics were prescribed at discharge (Figure 4).
- Line 123: Which antibiotics were used, and for how many days?
Response:
Thank you for this comment. We acknowledge that the original description regarding post-discharge antibiotic use was unclear. The cat had been managed at a local veterinary clinic following discharge, but specific information on the antibiotics used was not available to us. To avoid confusion and ensure clarity, we have revised the sentence to a more concise version that focuses on the patient's clinical status at the time of referral:
“Two months later, the cat was referred back to our hospital with worsening clinical signs, including cough, dyspnea, and lethargy.”
This revised sentence reflects the information that was verifiable from our records.
Line 137-139
Two months later, the cat was referred back to our hospital with worsening clinical signs, including cough, dyspnea, and lethargy.
- Line 143: Which antibiotic therapy are you referring to? It is unclear. Also, on what basis was meropenem chosen?
Response:
Thank you for your comment. The term “initial antibiotic therapy” refers to the previously used combination of amoxicillin-clavulanate, marbofloxacin, and metronidazole. During the recurrence, this regimen was found to be ineffective. Based on the previous culture and susceptibility test results, metronidazole was replaced with meropenem (10 mg/kg IV q8h) to enhance anaerobic coverage and address potential resistance. We have revised the manuscript to clarify this change and its rationale.
Line 159-165
As the initial antibiotic therapy—amoxicillin-clavulanate, marbofloxacin, and metronidazole—proved ineffective during the recurrent episode, metronidazole was replaced with meropenem (10 mg/kg IV q8h) based on the previous culture and susceptibility results to enhance anaerobic coverage and address potential resistance. In addition, meloxicam (0.2 mg/kg SC q24h) and mirtazapine (1.88 mg/cat PO q48h) were initially administered to reduce fever and stimulate appetite, respectively, but both were discontinued as clinical signs improved.
- Line 150: Was a culture performed on the pleural effusion?
Response:
Thank you for the question. Yes, a bacterial culture was performed on the pleural effusion collected during thoracotomy. We have revised the manuscript to clearly state that culture testing was conducted at this time point
Line 178-180
The pleural fluid obtained was mildly turbid, yellow, and had low viscosity (Figure 7C). A sample of the fluid was subjected to bacterial culture and antibiotic susceptibility. (Neodin BioVet Laboratory, Seoul, Korea).
- Line 170: When was this culture performed and on which sample? It is unclear; please rewrite.
Response:
Thank you for your comment. As noted in our response to Comment 9, the bacterial culture was performed on pleural fluid collected intraoperatively during thoracotomy. The culture result—identifying Burkholderia cenocepacia—was reported at this stage. We have revised the manuscript to clarify the timing and source of the sample accordingly
Line 203-206
The bacterial culture result from the sample collected intraoperatively during thoracotomy was reported at this time. Based on the results of bacterial culture (Burkholderia cenocepacia) and susceptibility testing, antibiotic therapy was unchanged.
- Line 174: So, was corticosteroid added while NSAIDs were still being administered? This is unclear.
Response:
Thank you for your comment. As described in the revised text related to Comment no. 8, meloxicam (NSAID) was discontinued prior to the initiation of corticosteroid therapy. Therefore, NSAIDs and corticosteroids were not administered concurrently. We have ensured this timeline is clearly reflected in the revised manuscript.
- I would also like to know the bacterial count of the cultures.
Response:
Thank you for your question. Bacterial identification was performed using Bruker MALDI-TOF mass spectrometry at Neodin Biolab. Unfortunately, the laboratory did not report quantitative bacterial counts for the cultures.
- In the discussion, the hypothesis that a 3-week antibiotic treatment might have been insufficient—which could have contributed to the formation of the empyema—was not considered.
Response:
Thank you for your valuable suggestion. We have revised the discussion section to address the possibility that recurrence could have resulted from insufficient antibiotic duration. However, considering that the initial episode responded well to therapy and no additional infectious agents were identified upon recurrence, and that the cultured organism (Burkholderia cenocepacia) demonstrated in vitro susceptibility with limited clinical response, we concluded that recurrence was unlikely to be caused by inadequate antibiotic coverage during the initial episode. This reasoning has now been clearly stated in the revised manuscript
Line 246-258
At the time of recurrence, the clinical presentation had progressed to caudal mediastinal paraesophageal empyema (CMPE), which differed from the initial acute and severe septic pyothorax that had responded rapidly to antibiotic therapy. Despite the isolation of Burkholderia cenocepacia on culture and its demonstrated susceptibility to the administered antibiotics, the patient's clinical condition continued to deteriorate. In contrast, previous studies have reported successful management of recurrent feline pyothorax using the same antibiotics employed in the initial treatment, highlighting a discrepancy with the present case [3]. In contrast, studies in dogs have reported poor prognosis and high mortality rates associated with recurrence [9].
Given that no other infectious organisms were identified through cytological or histopathological examination, and that even the cultured bacteria showed in vitro susceptibility to the treatment regimen with minimal clinical improvement, it was considered unlikely that the recurrence was due to insufficient antibiotic coverage during the initial episode.
- Line 251: It states that ventricular thickening resolved after 2 months, but in line 179, it is mentioned that the follow-up was done after 6 months. Please rewrite to clarify the timeline.
Response:
Thank you for pointing out this discrepancy. We have clarified the timeline in the revised manuscript. The resolution of ventricular thickening and discontinuation of cardiac medications occurred at the 2-month follow-up. Thereafter, the patient remained clinically stable, and thoracic imaging and echocardiography performed at the 6-month follow-up confirmed the continued absence of abnormalities. This distinction between the resolution point and the total follow-up duration has been clearly reflected in the updated text.
Line 165-167
Echocardiography indicated the resolution of cardiac abnormalities, allowing the discontinuation of cardiac medications. Clinical deterioration prompted a thoracotomy 12 days after recurrence.

Round 2
Reviewer 1 Report
Comments and Suggestions for Authors
Manuscript with ID: animals- 3585576 and Title: Successful Management of Recurrent Pyothorax in a Cat: Clinical Findings with Medical and Surgical Approaches
This report presents a case of medical and surgical management of recurrent pyothorax in a 10-year-old cat, highlighting the clinical findings, diagnostic assessments, and treatment strategies.
This article deals with a very interesting topic for clinical practice and I appreciate the work done by the authors. The manuscript is amended and revised. In my opinion it is suitable for publication.
Reviewer 3 Report
Comments and Suggestions for Authors
The authors improved the readability and now its read to be published